# It is all in the noise: Efficient multi-task Gaussian process inference with structured residuals

**Barbara Rakitsch**
Machine Learning and Computational Biology
Research Group
Max Planck Institutes Tübingen, Germany
rakitsch@tuebingen.mpg.de

**Christoph Lippert**
Microsoft Research
Los Angeles, USA
lippert@microsoft.com

**Karsten Borgwardt**[1,2]
Machine Learning and Computational Biology
Research Group
Max Planck Institutes Tübingen, Germany
karsten.borgwardt@tuebingen.mpg.de

**Oliver Stegle**[2]
European Molecular Biology Laboratory
European Bioinformatics Institute
Cambridge, UK
oliver.stegle@ebi.ac.uk

## Abstract

Multi-task prediction methods are widely used to couple regressors or classification models by sharing information across related tasks. We propose a multi-task Gaussian process approach for modeling both the relatedness between regressors and the task correlations in the residuals, in order to more accurately identify true sharing between regressors. The resulting Gaussian model has a covariance term in form of a sum of Kronecker products, for which efficient parameter inference and out of sample prediction are feasible. On both synthetic examples and applications to phenotype prediction in genetics, we find substantial benefits of modeling structured noise compared to established alternatives.

## 1  Introduction

Multi-task Gaussian process (GP) models are widely used to couple related tasks or functions for joint regression. This coupling is achieved by designing a structured covariance function, yielding a prior on vector-valued functions. An important class of structured covariance functions can be derived from a product of a kernel function $c$ relating the tasks (task covariance) and a kernel function $r$ relation the samples (sample covariance)

$$\text{cov}(f_{n,t}, f_{n',t'}) = \underbrace{c(t,t')}_{\text{task covariance}} \cdot \underbrace{r(n,n')}_{\text{sample covariance}} \ , \tag{1}$$

where $f_{n,t}$ are latent function values that induce the outputs $y_{n,t}$ by adding some Gaussian noise. If the outputs $y_{n,t}$ are fully observed, with one training example per sample and task, the resulting covariance matrix between the latent factors can be written as a Kronecker product between the sample covariance matrix and the task covariance matrix (e.g. [1]). More complex multi-task covariance structures can be derived from generalizations of this product structure, for example via convolution of multiple features, e.g. [2]. In [3], a parameterized covariance over the tasks is used, assuming that task-relevant features are observed. The authors of [4] couple the latent features over the tasks exploiting a dependency in neural population activity over time.

Work proposing this type of multi-task GP regression builds on Bonilla and Williams [1], who have emphasized that the power of Kronecker covariance models for GP models (Eqn. (1)) is linked to non-zero observation noise. In fact, in the limit of noise-free training observations, the coupling of tasks for predictions is lost in the predictive model, reducing to ordinary GP regressors for each individual task. Most multi task GP models build on a simple independent noise model, an assumption that is mainly routed in computational convenience. For example [5] show that this assumption renders the evaluation of the model likelihood and parameter gradients tractable, avoiding the explicit evaluation of the Kronecker covariance.

In this paper, we account for residual noise structure by modeling the signal and the noise covariance matrix as two separate Kronecker products. The structured noise covariance is independent of the inputs but instead allows to capture residual correlation between tasks due to latent causes; moreover, the model is simple and extends the widely used product covariance structure. Conceptually related noise models have been proposed in animal breeding [6, 7]. In geostatistics [8], linear coregionalization models have been introduced to allow for more complicated covariance structures: the signal covariance matrix is modeled as a sum of Kronecker products and the noise covariance as a single Kronecker product. In machine learning, the Gaussian process regression networks [9] considers an adaptive mixture of GPs to model related tasks. The mixing coefficients are dependent on the input signal and control the signal and noise correlation simultaneously.

The remainder of this paper is structured as follows. First, we show that unobserved regressors or causal processes inevitably lead to correlated residual, motivating the need to account for structured noise (Section 2). This extension of the multi task GP model allows for more accurate estimation of the task-task relationships, thereby improving the performance for out-of-sample predictions. At the same time, we show how an efficient inference scheme can be derived for this class of models. The proposed implementation handles closed form marginal likelihoods and parameter gradients for matrix-variate normal models with a covariance structure represented by the sum of two Kronecker products. These operations can be implemented at marginal extra computational cost compared to models that ignore residual task correlations (Section 3). In contrast to existing work extending Gaussian process multi task models by defining more complex covariance structures [2, 9, 8], our model utilizes the gradient of the marginal likelihood for parameter estimation and does not require expected maximization, variational approximation or MCMC sampling. We apply the resulting model in simulations and real settings, showing that correlated residuals are a concern in important applications (Section 4).

## 2 Multi-task Gaussian processes with structured noise

Let $\mathbf{Y} \in \mathbb{R}^{N \times T}$ denote the $N \times T$ output training matrix for $N$ samples and $T$ tasks. A column of this matrix corresponds to a particular task $t$ is denoted as $\mathbf{y}_t$, and $\mathrm{vec}\mathbf{Y} = \left(\mathbf{y}_1^\top \ldots \mathbf{y}_T^\top\right)^\top$ denotes the vector obtained by vertical concatenation of all columns of $\mathbf{Y}$. We indicate the dimensions of the matrix as capital subscripts when needed for clarity. A more thoughtful derivation of all equations can be found in the Supplementary Material.

**Multivariate linear model equivalence**  The multi-task Gaussian process regression model with structured noise can be derived from the perspective of a linear multivariate generative model. For a particular task $t$, the outputs are determined by a linear function of the training inputs across $F$ features $\mathbf{S} = \{\mathbf{s}_1, \ldots, \mathbf{s}_F\}$,

$$\mathbf{y}_t = \sum_{f=1}^{F} \mathbf{s}_f w_{f,t} + \boldsymbol{\psi}_t. \tag{2}$$

Multi-task sharing is achieved by specifying a multivariate normal prior across tasks, both for the regression weights $w_{f,t}$ and the noise variances $\boldsymbol{\psi}_t$:

$$p(\mathbf{W}^\top) = \prod_{f=1}^{F} \mathcal{N}\left(\mathbf{w}_f \,|\, \mathbf{0}, \mathbf{C}_{TT}\right) \quad p(\boldsymbol{\Psi}^\top) = \prod_{n=1}^{N} \mathcal{N}\left(\boldsymbol{\psi}_n \,|\, \mathbf{0}, \boldsymbol{\Sigma}_{TT}\right).$$

Marginalizing out the weights $\mathbf{W}$ and the residuals $\mathbf{\Psi}$ results in a matrix-variate normal model with sum of Kronecker products covariance structure

$$p(\text{vec}\mathbf{Y}\mid \mathbf{C},\mathbf{R},\mathbf{\Sigma}) = \mathcal{N}\left(\text{vec}\mathbf{Y}_{NT}\mid \mathbf{0}, \underbrace{\mathbf{C}_{TT}\otimes\mathbf{R}_{NN}}_{\text{signal covariance}} + \underbrace{\mathbf{\Sigma}_{TT}\otimes\mathbf{I}_{NN}}_{\text{noise covariance}}\right), \tag{3}$$

where $\mathbf{R}_{NN} = \mathbf{S}\mathbf{S}^{\top}$ is the sample covariance matrix that results from the marginalization over the weights $\mathbf{W}$ in Eqn. (2). In the following, we will refer to a Gaussian process model with this type of sum of Kronecker products covariance structure as GP-kronsum[1]. As common to any kernel method, the linear covariance $\mathbf{R}$ can be replaced with any positive semi-definite covariance function.

**Predictive distribution**  In a GP-kronsum model, predictions for unseen test instances can be carried out by using the standard Gaussian process framework [10]:

$$p(\text{vec}\mathbf{Y}^*\mid\mathbf{R}^*,\mathbf{Y}) = \mathcal{N}\left(\text{vec}\mathbf{Y}^*\mid\text{vec}\,\mathbf{M}^*,\mathbf{V}^*\right). \tag{4}$$

Here, $\mathbf{M}^*$ denotes the mean prediction and $\mathbf{V}^*$ is the predictive covariance. Analytical expression for both can be obtained by considering the joint distribution of observed and unobserved outputs and completing the square, yielding:

$$\text{vec}\,\mathbf{M}^* = \left(\mathbf{C}_{TT}\otimes\mathbf{R}^*_{N^*N}\right)\left(\mathbf{C}_{TT}\otimes\mathbf{R}_{NN} + \mathbf{\Sigma}_{TT}\otimes\mathbf{I}_{NN}\right)^{-1}\text{vec}\mathbf{Y}_{NT},$$

$$\mathbf{V}^* = \left(\mathbf{C}_{TT}\otimes\mathbf{R}^*_{N^*N^*}\right) - \left(\mathbf{C}_{TT}\otimes\mathbf{R}^*_{N^*N}\right)\left(\mathbf{C}_{TT}\otimes\mathbf{R}_{NN} + \mathbf{\Sigma}_{TT}\otimes\mathbf{I}_{NN}\right)^{-1}\left(\mathbf{C}_{TT}\otimes\mathbf{R}^*_{NN^*}\right),$$

where $\mathbf{R}^*_{N^*N}$ is the covariance matrix between the test and training instances, and $\mathbf{R}^*_{N^*N^*}$ is the covariance matrix between the test samples.

**Design of multi-task covariance function**  In practice, neither the form of $\mathbf{C}$ nor the form of $\mathbf{\Sigma}$ is known *a priori* and hence needs to be inferred from data, fitting a set of corresponding covariance parameters $\theta_C$ and $\theta_\Sigma$. If the number of tasks $T$ is large, learning a free-form covariance matrix is prone to overfitting, as the number of free parameters grows quadratically with $T$. In the experiments, we consider a rank-$k$ approximation of the form $\sum_{k=1}^{K}\mathbf{x}_k\mathbf{x}_k^{\top} + \sigma^2\mathbf{I}$ for the task matrices.

**Task cancellation when the task covariance matrices are equal**  A notable form of the predictive distribution (4) arises for the special case $\mathbf{C} = \mathbf{\Sigma}$, that is the task covariance matrix of signal and noise are identical. Similar to previous results for noise-free observations [1], maximizing the marginal likelihood $p(\text{vec}\mathbf{Y}|\mathbf{C},\mathbf{R},\mathbf{\Sigma})$ with respect to the parameters $\theta_R$ becomes independent of $\mathbf{C}$ and the predictions are decoupled across tasks, i.e. the benefits from joint modeling are lost:

$$\text{vec}\,\mathbf{M}^* = \text{vec}\left(\mathbf{R}^*_{N^*N}(\mathbf{R}_{NN} + \mathbf{I}_{NN})^{-1}\mathbf{Y}_{NT}\right) \tag{5}$$

In this case, the predictions depend on the sample covariance, but not on the task covariance. Thus, the GP-kronsum model is most useful when the task covariances on observed features and on noise reflect two independent sharing structures.

## 3  Efficient Inference

In general, efficient inference can be carried out for Gaussian models with a sum covariance of two arbitrary Kronecker products

$$p(\text{vec}\mathbf{Y}\mid\mathbf{C},\mathbf{R},\mathbf{\Sigma}) = \mathcal{N}\left(\text{vec}\mathbf{Y}\mid\mathbf{0},\mathbf{C}_{TT}\otimes\mathbf{R}_{NN} + \mathbf{\Sigma}_{TT}\otimes\mathbf{\Omega}_{NN}\right). \tag{6}$$

The key idea is to first consider a suitable data transformation that leads to a diagonalization of all covariance matrices and second to exploit Kronecker tricks whenever possible.

Let $\mathbf{\Sigma} = \mathbf{U}_\Sigma\mathbf{S}_\Sigma\mathbf{U}_\Sigma^{\top}$ be the eigenvalue decomposition of $\mathbf{\Sigma}$, and analogously for $\mathbf{\Omega}$. Borrowing ideas from [11], we can first bring the covariance matrix in a more amenable form by factoring out the structured noise:

$$\mathbf{K} = \mathbf{C} \otimes \mathbf{R} + \mathbf{\Sigma} \otimes \mathbf{\Omega}$$
$$= \left(\mathbf{U}_\Sigma \mathbf{S}_\Sigma^{\frac{1}{2}} \otimes \mathbf{U}_\Omega \mathbf{S}_\Omega^{\frac{1}{2}}\right)\left(\tilde{\mathbf{C}} \otimes \tilde{\mathbf{R}} + \mathbf{I} \otimes \mathbf{I}\right)\left(\mathbf{S}_\Sigma^{\frac{1}{2}}\mathbf{U}_\Sigma^\top \otimes \mathbf{S}_\Omega^{\frac{1}{2}}\mathbf{U}_\Omega^\top\right), \tag{7}$$

where $\tilde{\mathbf{C}} = \mathbf{S}_\Sigma^{-\frac{1}{2}}\mathbf{U}_\Sigma^\top \mathbf{C} \mathbf{U}_\Sigma \mathbf{S}_\Sigma^{-\frac{1}{2}}$ and $\tilde{\mathbf{R}} = \mathbf{S}_\Omega^{-\frac{1}{2}}\mathbf{U}_\Omega^\top \mathbf{R} \mathbf{U}_\Omega \mathbf{S}_\Omega^{-\frac{1}{2}}$. In the following, we use definition $\tilde{\mathbf{K}} = \tilde{\mathbf{C}} \otimes \tilde{\mathbf{R}} + \mathbf{I} \otimes \mathbf{I}$ for this transformed covariance.

**Efficient log likelihood evaluation.** The log model likelihood (Eqn. (6)) can be expressed in terms of the transformed covariance $\tilde{\mathbf{K}}$:

$$\mathcal{L} = -\frac{NT}{2}\ln(2\pi) - \frac{1}{2}\ln|\mathbf{K}| - \frac{1}{2}\text{vec}\mathbf{Y}^\top \mathbf{K}^{-1}\text{vec}\mathbf{Y}$$
$$= -\frac{NT}{2}\ln(2\pi) - \frac{1}{2}\ln|\tilde{\mathbf{K}}| - \frac{1}{2}|\mathbf{S}_\Sigma \otimes \mathbf{S}_\Omega| - \frac{1}{2}\text{vec}\tilde{\mathbf{Y}}^\top \tilde{\mathbf{K}}^{-1}\text{vec}\tilde{\mathbf{Y}}, \tag{8}$$

where $\text{vec}\tilde{\mathbf{Y}} = \left(\mathbf{S}_\Sigma^{-\frac{1}{2}}\mathbf{U}_\Sigma^\top \otimes \mathbf{S}_\Omega^{-\frac{1}{2}}\mathbf{U}_\Omega^\top\right)\text{vec}\mathbf{Y} = \text{vec}\left(\mathbf{S}_\Omega^{-\frac{1}{2}}\mathbf{U}_\Omega^T \mathbf{Y} \mathbf{U}_\Sigma \mathbf{S}_\Sigma^{-\frac{1}{2}}\right)$ is the projected output. Except for the additional term $|\mathbf{S}_\Sigma \otimes \mathbf{S}_\Omega|$, resulting from the transformation, the log likelihood has the exactly same form as for multi-task GP regression with iid noise [1, 5]. Using an analogous derivation, we can now efficiently evaluate the log likelihood:

$$\mathcal{L} = -\frac{NT}{2}\ln(2\pi) - \frac{1}{2}\ln|\mathbf{S}_{\tilde{\mathbf{C}}} \otimes \mathbf{S}_{\tilde{\mathbf{R}}} + \mathbf{I} \otimes \mathbf{I}| - \frac{N}{2}\ln|\mathbf{S}_\Sigma| - \frac{T}{2}|\mathbf{S}_\Omega|$$
$$- \frac{1}{2}\text{vec}\left(\mathbf{U}_{\tilde{\mathbf{R}}}^\top \tilde{\mathbf{Y}} \mathbf{U}_{\tilde{\mathbf{C}}}\right)^\top (\mathbf{S}_{\tilde{\mathbf{C}}} \otimes \mathbf{S}_{\tilde{\mathbf{R}}} + \mathbf{I} \otimes \mathbf{I})^{-1}\text{vec}\left(\mathbf{U}_{\tilde{\mathbf{R}}}^\top \tilde{\mathbf{Y}} \mathbf{U}_{\tilde{\mathbf{C}}}\right), \tag{9}$$

where we have defined the eigenvalue decomposition of $\tilde{\mathbf{C}}$ as $\mathbf{U}_{\tilde{\mathbf{C}}}\mathbf{S}_{\tilde{\mathbf{C}}}\mathbf{U}_{\tilde{\mathbf{C}}}^\top$ and similar for $\tilde{\mathbf{R}}$.

**Efficient gradient evaluation** The derivative of the log marginal likelihood with respect to a covariance parameter $\theta_R$ can be expressed as:

$$\frac{\partial}{\partial \theta_R}\mathcal{L} = -\frac{1}{2}\frac{\partial}{\partial \theta_R}\ln|\tilde{\mathbf{K}}| - \frac{1}{2}\text{vec}\tilde{\mathbf{Y}}^\top\left(\frac{\partial}{\partial \theta_R}\tilde{\mathbf{K}}^{-1}\right)\text{vec}(\tilde{\mathbf{Y}})$$
$$= -\frac{1}{2}\text{diag}\left((\mathbf{S}_{\tilde{\mathbf{C}}} \otimes \mathbf{S}_{\tilde{\mathbf{R}}} + \mathbf{I} \otimes \mathbf{I})^{-1}\right)^\top \text{diag}\left(\mathbf{S}_{\tilde{\mathbf{C}}} \otimes \mathbf{U}_{\tilde{\mathbf{R}}}^\top\left(\frac{\partial}{\partial \theta_R}\tilde{\mathbf{R}}\right)\mathbf{U}_{\tilde{\mathbf{R}}}\right)$$
$$+ \frac{1}{2}\text{vec}(\hat{\mathbf{Y}})^\top\text{vec}\left(\mathbf{U}_{\tilde{\mathbf{R}}}^\top\left(\frac{\partial}{\partial \theta_R}\tilde{\mathbf{R}}\right)\mathbf{U}_{\tilde{\mathbf{R}}}\hat{\mathbf{Y}}\mathbf{S}_{\tilde{\mathbf{C}}}\right), \tag{10}$$

where $\text{vec}(\hat{\mathbf{Y}}) = (\mathbf{S}_{\tilde{\mathbf{C}}} \otimes \mathbf{S}_{\tilde{\mathbf{R}}} + \mathbf{I} \otimes \mathbf{I})^{-1}\text{vec}\left(\mathbf{U}_{\tilde{\mathbf{R}}}^\top \tilde{\mathbf{Y}} \mathbf{U}_{\tilde{\mathbf{C}}}\right)$. Analogous gradients can be derived for the task covariance parameters $\theta_C$ and $\theta_\Sigma$. The proposed speed-ups also apply to the special cases where $\mathbf{\Sigma}$ is modeled as being diagonal as in [1], or for optimizing the parameters of a kernel function. Since the sum of Kronecker products generally can not be written as a single Kronecker product, the speed-ups cannot be generalized to larger sums of Kronecker products.

**Efficient prediction** Similarly, the mean predictor (Eqn. (4)) can be efficiently evaluated

$$\text{vec}\,\mathbf{M}^* = \text{vec}\left[\left(\mathbf{R}^*\mathbf{U}_\Omega \mathbf{S}_\Omega^{-\frac{1}{2}}\right)\left(\mathbf{U}_{\tilde{\mathbf{R}}}\hat{\mathbf{Y}}\mathbf{U}_{\tilde{\mathbf{C}}}^\top\right)\left(\mathbf{S}_\Sigma^{-\frac{1}{2}}\mathbf{U}_\Sigma^\top \mathbf{C}^\top\right)\right]. \tag{11}$$

**Gradient-based parameter inference** The closed-form expression of the marginal likelihood (Eqn. (9)) and gradients with respect to covariance parameters (Eqn. (10)) allow for use of gradient-based parameter inference. In the experiments, we employ a variant of L-BFGS-B [12].

**Computational cost.** While the naive approach has a runtime of $O(N^3 \cdot T^3)$ and memory requirement of $O(N^2 \cdot T^2)$, as it explicitly computes and inverts the Kronecker products, our reformulation reduces the runtime to $O(N^3 + T^3)$ and the memory requirement to $O(N^2 + T^2)$, making it applicable to large numbers of samples and tasks. The empirical runtime savings over the naive approach are explored in Section 4.1.

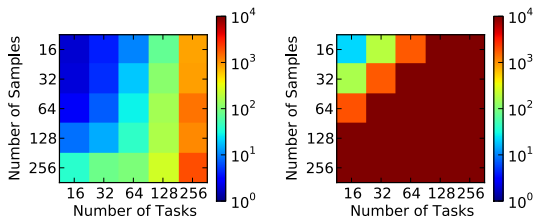

Figure 1: **Runtime comparison on synthetic data.** We compare our efficient GP-kronsum implementation (left) versus its naive counterpart (right). Shown is the runtime in seconds on a logarithmic scale as a function of the sample size and the number of tasks. The optimization was stopped prematurely if it did not complete after $10^4$ seconds.

(a) Efficient Implementation    (b) Naive Implementation

## 4  Experiments

We investigated the performance of the proposed GP-kronsum model in both simulated datasets and response prediction problems in statistical genetics. To investigate the benefits of structured residual covariances, we compared the GP-kronsum model to a Gaussian process (GP-kronprod) with iid noise [5] as well as independent modeling of tasks using a standard Gaussian process (GP-single), and joint modeling of all tasks using a standard Gaussian on a pooled dataset, naively merging data from all tasks (GP-pool).

The predictive performance of individual models was assessed through 10-fold cross-validation. For each fold, model parameters were fit on the training data only. To avoid local optima during training, parameter fitting was carried out using five random restarts of the parameters on 90% of the training instances. The remaining 10% of the training instances were used for out of sample selection using the maximum log likelihood as criterion. Unless stated otherwise, in the multi-task models the relationship between tasks was parameterized as $\mathbf{x}\mathbf{x}^\top + \sigma^2\mathbf{I}$, the sum of a rank-1 matrix and a constant diagonal component. Both parameters, $\mathbf{x}$ and $\sigma^2$, were learnt by optimizing the marginal likelihood. Finally, we measured the predictive performance of the different methods via the averaged square of Pearson's correlation coefficient $r^2$ between the true and the predicted output, averaged over tasks. The squared correlation coefficient is commonly used in statistical genetics to evaluate the performance of different predictors [13].

### 4.1  Simulations

First, we considered simulated experiments to explore the runtime behavior and to find out if there are settings in which GP-kronsum performs better than existing methods.

**Runtime evaluation.** As a first experiment, we examined the runtime behavior of our method as a function of the number of samples and of the number of tasks. Both parameters were varied in the range $\{16, 32, 64, 128, 256\}$. The simulated dataset was drawn from the GP-kronsum model (Eqn. (3)) using a linear kernel for the sample covariance matrix $\mathbf{R}$ and rank-1 matrices for the task covariances $\mathbf{C}$ and $\mathbf{\Sigma}$. The runtime of this model was assessed for a single likelihood optimization on an AMD Opteron Processor 6,378 using a single core (2.4GHz, 2,048 KB Cache, 512 GB Memory) and compared to a naive implementation. The optimization was stopped prematurely if it did not converge within $10^4$ seconds.

In the experiments, we used a standard linear kernel on the features of the samples as sample covariance while learning the task covariances. This modeling choice results in a steeper runtime increase with the number of tasks, due to the increasing number of model parameters to be estimated. Figure 1 demonstrates the significant speed-up. While our algorithm can handle 256 samples/256 tasks with ease, the naive implementation failed to process more than 32 samples/32 tasks.

**Unobserved causal process induces structured noise** A common source of structured residuals are unobserved causal processes that are not captured via the inputs. To explore this setting, we generated simulated outputs from a sum of two different processes. For one of the processes, we assumed that the causal features $\mathbf{X}_{\text{obs}}$ were observed, whereas for the second process the causal features $\mathbf{X}_{\text{hidden}}$ were hidden and independent of the observed measurements. Both processes were simulated to have a linear effect on the output. The effect from the observed features was again divided up into an independent effect, which is task-specific, and a common effect, which, up to

rescaling $\mathbf{r}_{common}$, is shared over all tasks:

$$\mathbf{Y}_{\text{common}} = \mathbf{X}_{\text{obs}}\mathbf{W}_{\text{common}}, \ \mathbf{W}_{\text{common}} = \mathbf{r}_{\text{common}} \otimes \mathbf{w}_{\text{common}}, \ \mathbf{r}_{\text{common}} \sim \mathcal{N}(\mathbf{0}, \mathbf{I}), \mathbf{w}_{\text{common}} \sim \mathcal{N}(\mathbf{0}, \mathbf{I})$$

The trade-off parameter $\mu_{\text{common}}$ determines the extent of relatedness between tasks:

$$\mathbf{Y}_{\text{obs}} = \mu_{\text{common}}\mathbf{Y}_{\text{common}} + (1 - \mu_{\text{common}})\mathbf{Y}_{\text{ind}}.$$

The effect of the hidden features was simulated analogously. A second trade-off parameter $\mu_{\text{hidden}}$ was introduced, controlling the ratio between the observed and hidden effect:

$$\mathbf{Y} = \mu_{\text{signal}} \left[ (1 - \mu_{\text{hidden}})\mathbf{Y}_{\text{obs}} + \mu_{\text{hidden}}\mathbf{Y}_{\text{hidden}} \right] + (1 - \mu_{\text{signal}})\mathbf{Y}_{\text{noise}},$$

where $\mathbf{Y}_{\text{noise}}$ is Gaussian observation noise, and $\mu_{\text{signal}}$ is a third trade-off parameter defining the ratio between noise and signal.

To investigate the impact of the different trade-off parameters, we considered a series of datasets varying one of the parameters while keeping others fixed. We varied $\mu_{\text{signal}}$ in the range $\{0.1, 0.3, 0.5, 0.7, \mathbf{0.9}, 1.0\}$, $\mu_{\text{common}} \in \{0.0, 0.1, 0.3, 0.5, 0.7, \mathbf{0.9}, 1.0\}$ and $\mu_{\text{hidden}} \in \{0.0, 0.1, 0.3, \mathbf{0.5}, 0.7, 0.9, 1.0\}$, with default values marked in bold. Note that the best possible explained variance for the default setting is $45\%$, as the causal signal is split up equally between the observed and the hidden process. For all simulation experiments, we created datasets with $200$ samples and 10 tasks. The number of observed features was set to $200$, as well as the number of hidden features. For each such simulation setting, we created 30 datasets.

First, we considered the impact of variation in signal strength $\mu_{\text{signal}}$ (Figure 2a), where the overall signal was divided up equally between the observed and hidden signal. Both GP-single and GP-kronsum performed better as the overall signal strength increased. The performance of GP-kronsum was superior, as the model can exploit the relatedness between the different tasks.
Second, we explored the ability of the different methods to cope with an underlying hidden process (Figure 2b). In the absence of a hidden process ($\mu_{\text{hidden}} = 0$), GP-kronprod and GP-kronsum had very similar performances, as both methods leverage the shared signal of the observed process, thereby outperforming the single-task GPs. However, as the magnitude of the hidden signal increases, GP-kronprod falsely explains the task correlation completely by the covariance term representing the observed process which leads to loss of predictive power.
Last, we examined the ability of different methods to exploit the relatedness between the tasks (Figure 2c). Since GP-single assumed independent tasks, the model performed very similarly across the full range of common signal. GP-kronprod suffered from the same limitations as previously described, because the correlation between tasks in the hidden process increases synchronously with the correlation in the observed process as $\mu_{\text{common}}$ increases. In contrast, GP-kronsum could take advantage of the shared component between the tasks, as knowledge is transferred between them.
GP-pool was consistently outperformed by all competitors as two of its main assumptions are heavily violated: samples of different tasks do not share the same signal and the residuals are neither independent of each other, nor do they have the same noise level.

In summary, the proposed model is robust to a range of different settings and clearly outperforms its competitors when the tasks are related to each other and not all causal processes are observed.

## 4.2 Applications to phenotype prediction

As a real world application we considered phenotype prediction in statistical genetics. The aim of these experiments was to demonstrate the relevance of unobserved causes in real world prediction problems and hence warrant greater attention.

**Gene expression prediction in yeast** We considered gene expression levels from a yeast genetics study [14]. The dataset comprised of gene expression levels of $5,493$ genes and $2,956$ SNPs (features), measured for 109 yeast crosses. Expression levels for each cross were measured in two conditions (glucose and ethanol as carbon source), yielding a total of 218 samples. In this experiment, we treated the condition information as a hidden factor instead of regressing it out, which is analogous to the hidden process in the simulation experiments. The goal of this experiment was to investigate how alternative methods can deal and correct for this hidden covariate. We normalized all features and all tasks to zero mean and unit variance. Subsequently, we filtered out all genes that were not consistently expressed in at least 90% of the samples ($z$-score cutoff 1.5). We also

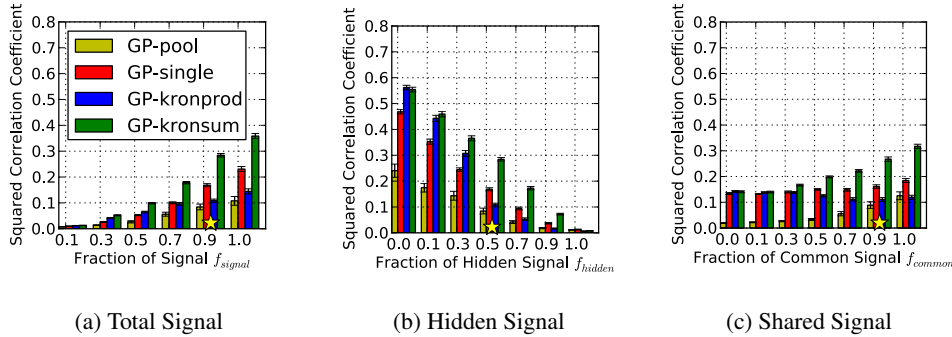

|   (a) Total Signal   |   (b) Hidden Signal   |   (c) Shared Signal   |

Figure 2: **Evaluation of alternative methods for different simulation settings.** From left to right: (a) Evaluation for varying signal strength. (b) Evaluation for variable impact of the hidden signal. (c) Evaluation for different strength of relatedness between the tasks. In each simulation setting, all other parameters were kept constant at default parameters marked with the yellow star symbol.

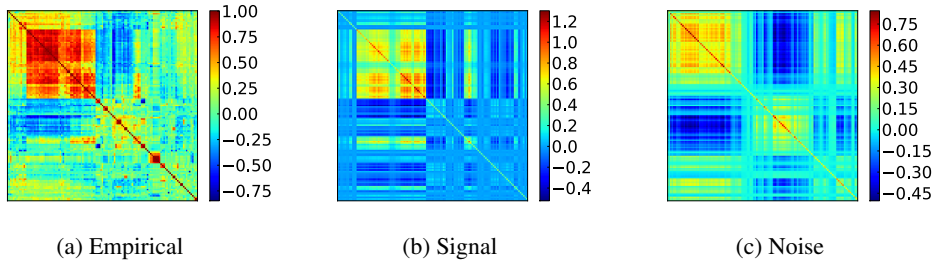

|   (a) Empirical   |   (b) Signal   |   (c) Noise   |

Figure 3: **Fitted task covariance matrices for gene expression levels in yeast.** From left to right: (a) Empirical covariance matrix of the gene expression levels. (b) Signal covariance matrix learnt by GP-kronsum. (c) Noise covariance matrix learnt by GP-kronsum. The ordering of the tasks was determined using hierarchical clustering on the empirical covariance matrix.

discarded genes with low signal ($< 10\%$ of the variance) or were close to noise free ($> 90\%$ of the variance), reducing the number of genes to 123, which we considered as tasks in our experiment. The signal strength was estimated by a univariate GP model. We used a linear kernel calculated on the SNP features for the sample covariance.

Figure 3 shows the empirical covariance and the learnt task covariances by GP-kronsum. Both learnt covariances are highly structured, demonstrating that the assumption of iid noise in the GP-kronprod model is violated in this dataset. While the signal task covariance matrix reflects genetic signals that are shared between the gene expression levels, the noise covariance matrix mainly captures the mean shift between the two conditions the gene expression levels were measured in (Figure 4). To investigate the robustness of the reconstructed latent factor, we repeated the training 10 times. The mean latent factors and its standard errors were $0.2103 \pm 0.0088$ (averaged over factors, over the 10 best runs selected by out-of-sample likelihood), demonstrating robustness of the inference.

When considering alternative methods for out of sample prediction, the proposed Kronecker Sum model ($r^2$(GP-kronsum)$=0.3322 \pm 0.0014$) performed significantly better than previous approaches ($r^2$(GP-pool)$=0.0673 \pm 0.0004$, $r^2$(GP-single)$=0.2594 \pm 0.0011$, $r^2$(GP-kronprod)$=0.1820 \pm 0.0020$). The results are averages over 10 runs and $\pm$ denotes the corresponding standard errors.

**Multi-phenotype prediction in** *Arabidopsis thaliana*. As a second dataset, we considered a genome-wide association study in *Arabidopsis thaliana* [15] to assess the prediction of developmental phenotypes from genomic data. This dataset consisted of 147 samples and 216,130 single nucleotide polymorphisms (SNPs, here used as features). As different tasks, we considered the phenotypes *flowering period duration*, *life cycle period*, *maturation period* and *reproduction period*. To avoid outliers and issues due to non-Gaussianity, we preprocessed the phenotypic data by first converting it to ranks and squashing the ranks through the inverse cumulative Gaussian distribution. The SNPs in *Arabidopsis thaliana* are binary and we discarded features with a frequency of less

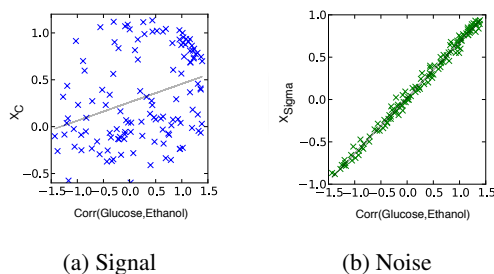

(a) Signal                    (b) Noise

Figure 4: **Correlation between the mean difference of the two conditions and the latent factors on the yeast dataset.** Shown is the strength of the latent factor of the signal (left) and the noise (right) task covariance matrix as a function of the mean difference between the two environmental conditions. Each dot corresponds to one gene expression level.

than 10% in all samples, resulting in 176,436 SNPs. Subsequently, we normalized the features to zero mean and unit variance. Again, we used a linear kernel on the SNPs as sample covariance.

Since the causal processes in *Arabidopsis thaliana* are complex, we allowed the rank of the signal and noise matrix to vary between 1 and 3. The appropriate rank complexity was selected on the 10% hold out data of the training fold. We considered the average squared correlation coefficient on the holdout fraction of the training data to select the model for prediction on the test dataset. Notably, for GP-kronprod, the selected task complexity was $\text{rank}(\mathbf{C}) = 3$, whereas GP-kronsum selected a simpler structure for the signal task covariance ($\text{rank}(\mathbf{C}) = 1$) and chose a more complex noise covariance, $\text{rank}(\mathbf{\Sigma}) = 2$.

The cross validation prediction performance of each model is shown in Table 1. For *reproduction period*, GP-single is outperformed by all other methods. For the phenotype *life cycle period*, the noise estimates of the univariate GP model were close to zero, and hence all methods, except of GP-pool, performed equally well since the measurements of the other phenotypes do not provide additional information. For *maturation period*, GP-kronsum and GP-kronprod showed improved performance compared to GP-single and GP-pool. For *flowering period duration*, GP-kronsum outperformed its competitors.

| | Flowering period duration | Life cycle period | Maturation period | Reproduction period |
|---|---|---|---|---|
| GP-pool | $0.0502 \pm 0.0025$ | $0.1038 \pm 0.0034$ | $0.0460 \pm 0.0024$ | $\mathbf{0.0478 \pm 0.0013}$ |
| GP-single | $0.0385 \pm 0.0017$ | $\mathbf{0.3500 \pm 0.0069}$ | $0.1612 \pm 0.0027$ | $0.0272 \pm 0.0024$ |
| GP-kronprod | $0.0846 \pm 0.0021$ | $\mathbf{0.3417 \pm 0.0062}$ | $\mathbf{0.1878 \pm 0.0042}$ | $\mathbf{0.0492 \pm 0.0032}$ |
| GP-kronsum | $\mathbf{0.1127 \pm 0.0049}$ | $\mathbf{0.3485 \pm 0.0068}$ | $\mathbf{0.1918 \pm 0.0041}$ | $\mathbf{0.0501 \pm 0.0033}$ |

Table 1: **Predictive performance of the different methods on the *Arabidopsis thaliana* dataset.** Shown is the squared correlation coefficient and its standard error (measured by repeating 10-fold cross-validation 10 times).

## 5  Discussion and conclusions

Multi-task Gaussian process models are a widely used tool in many application domains, ranging from the prediction of user preferences in collaborative filtering to the prediction of phenotypes in computational biology. Many of these prediction tasks are complex and important causal features may remain unobserved or are not modeled. Nevertheless, most approaches in common usage assume that the observation noise is independent between tasks. We here propose the GP-kronsum model, which allows to efficiently model data where the noise is dependent between tasks by building on a sum of Kronecker products covariance. In applications to statistical genetics, we have demonstrated (1) the advantages of the dependent noise model over an independent noise model, as well as (2) the feasibility of applying larger data sets by the efficient learning algorithm.

## Acknowledgement

We thank Francesco Paolo Casale for helpful discussions. OS was supported by an Marie Curie FP7 fellowship. KB was supported by the Alfried Krupp Prize for Young University Teachers of the Alfried Krupp von Bohlen und Halbach-Stiftung.

## Footnotes

[1] Also at Zentrum für Bioinformatik, Eberhard Karls Universität Tübingen,Tübingen, Germany

[2] Both authors contributed equally to this work.

[1]the covariance is defined as the sum of two Kronecker products and not as the classical Kronecker sum $\mathbf{C}\oplus\mathbf{R} = \mathbf{C}\otimes\mathbf{I} + \mathbf{I}\otimes\mathbf{R}$.

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
