[Supplementary Material · supplementary.pdf]

# It is all in the noise: Efficient multi-task Gaussian process inference with structured residuals

# Supplementary Material

**Barbara Rakitsch**
Machine Learning and Computational Biology
Research Group
Max Planck Institutes Tübingen, Germany
rakitsch@tuebingen.mpg.de

**Christoph Lippert**
Microsoft Research
Los Angeles, USA
lippert@microsoft.com

**Karsten Borgwardt**[1,2]
Machine Learning and Computational Biology
Research Group
Max Planck Institutes Tübingen, Germany
karsten.borgwardt@tuebingen.mpg.de

**Oliver Stegle**[2]
European Molecular Biology Laboratory
European Bioinformatics Institute
Cambridge, UK
oliver.stegle@ebi.ac.uk

## 1   Kronecker Identities

Let $\mathbf{A}$ be a $m \times n$ matrix, and $\mathbf{B}$ be a $p \times q$ matrix. The Kronecker product $\mathbf{A} \otimes \mathbf{B}$ is a $mp \times nq$ matrix and defined as follows:

$$\mathbf{A} \otimes \mathbf{B} = \begin{pmatrix} A_{11}\mathbf{B} & \dots & A_{1n}\mathbf{B} \\ \vdots & \ddots & \vdots \\ A_{m1}\mathbf{B} & \dots & A_{mn}\mathbf{B} \end{pmatrix} \tag{1}$$

The following equalities hold [1]:

$$
\begin{aligned}
(\mathbf{A} \otimes \mathbf{B})(\mathbf{C} \otimes \mathbf{D}) &= \mathbf{AC} \otimes \mathbf{BD} & (2) \\
(\mathbf{A} \otimes \mathbf{B})^{\top} &= \mathbf{A}^{\top} \otimes \mathbf{B}^{\top} & (3) \\
(\mathbf{A} \otimes \mathbf{B})^{-1} &= \mathbf{A}^{-1} \otimes \mathbf{B}^{-1} & (4) \\
|\mathbf{A} \otimes \mathbf{B}| &= |\mathbf{A}|^{p} \cdot |\mathbf{B}|^{n} & (5) \\
(\mathbf{A} \otimes \mathbf{B})\operatorname{vec}(\mathbf{Y}) &= \operatorname{vec}\left(\mathbf{BYA}^{\top}\right) & (6)
\end{aligned}
$$

Let $\mathbf{U}_A \mathbf{S}_A \mathbf{U}_A^{\top}$ be the eigenvalue decomposition of $\mathbf{A}$ and $\mathbf{U}_B \mathbf{S}_B \mathbf{U}_B^{\top}$ the eigenvalue decomposition of $\mathbf{B}$, then

$$\left(\mathbf{U}_A \otimes \mathbf{U}_B\right)\left(\mathbf{S}_A \otimes \mathbf{S}_B + \sigma^2 \mathbf{I}\right)\left(\mathbf{U}_A^{\top} \otimes \mathbf{U}_B^{\top}\right) \tag{7}$$

is the eigenvalue decomposition of $\mathbf{A} \otimes \mathbf{B} + \sigma^2 \mathbf{I}$, where $\sigma^2$ is a non-negative scalar.

## 1.1 Task Cancellation when the task covariance matrices are equal

For the special case $\mathbf{C} = \mathbf{\Sigma}$, the predictions become independent across tasks:

$$
\begin{aligned}
\mathrm{vec}\,\mathbf{M}^* &= (\mathbf{C} \otimes \mathbf{R}^*)(\mathbf{C} \otimes \mathbf{R} + \mathbf{\Sigma} \otimes \mathbf{I})^{-1}\,\mathrm{vec}\mathbf{Y} \\
&= (\mathbf{C} \otimes \mathbf{R}^*)(\mathbf{C} \otimes \mathbf{R} + \mathbf{C} \otimes \mathbf{I})^{-1}\,\mathrm{vec}\mathbf{Y} \\
&= (\mathbf{C} \otimes \mathbf{R}^*)(\mathbf{C} \otimes (\mathbf{R} + \mathbf{I}))^{-1}\,\mathrm{vec}\mathbf{Y} \\
&= (\mathbf{C} \otimes \mathbf{R}^*)\left(\mathbf{C}^{-1} \otimes (\mathbf{R} + \mathbf{I})^{-1}\right)\,\mathrm{vec}\mathbf{Y} \\
&= (\mathbf{C}\mathbf{C}^{-1} \otimes \mathbf{R}^*(\mathbf{R} + \mathbf{I})^{-1}\,\mathrm{vec}\mathbf{Y} \\
&= \left(\mathbf{I} \otimes \mathbf{R}^*(\mathbf{R} + \mathbf{I})^{-1}\right)\,\mathrm{vec}\mathbf{Y} \\
&= \mathrm{vec}\left(\mathbf{R}^*(\mathbf{R} + \mathbf{I})^{-1}\mathbf{Y}\right)
\end{aligned}
\tag{8}
$$

For the noiseless case $\mathbf{\Sigma} \to \mathbf{0}$, a similar proof can be obtained [2].

## 2 Efficient inference in matrix-variate sum of Kronecker products Gaussian models

In the following, we will show how to efficiently evaluate the marginal likelihood and parameter gradients for matrix-variate normal models with a sum of two Kronecker products as covariance.

**Efficient likelihood evaluation**     In a first step, we bring the covariance matrix in a more amenable form by factoring out the structured noise [3]:

$$
\begin{aligned}
\mathbf{K} &= \mathbf{C} \otimes \mathbf{R} + \mathbf{\Sigma} \otimes \mathbf{\Omega} \\
&= \mathbf{C} \otimes \mathbf{R} + \mathbf{U}_\Sigma \mathbf{S}_\Sigma \mathbf{U}_\Sigma^\top \otimes \mathbf{U}_\Omega \mathbf{S}_\Omega \mathbf{U}_\Omega^\top \\
&= \left(\mathbf{U}_\Sigma \mathbf{S}_\Sigma^{\frac{1}{2}} \otimes \mathbf{U}_\Omega \mathbf{S}_\Omega^{\frac{1}{2}}\right)\left(\mathbf{S}_\Sigma^{-\frac{1}{2}} \mathbf{U}_\Sigma^\top \mathbf{C} \mathbf{U}_\Sigma \mathbf{S}_\Sigma^{-\frac{1}{2}} \otimes \mathbf{S}_\Omega^{-\frac{1}{2}} \mathbf{U}_\Omega^\top \mathbf{R} \mathbf{U}_\Omega \mathbf{S}_\Omega^{-\frac{1}{2}} + \mathbf{I} \otimes \mathbf{I}\right)\left(\mathbf{S}_\Sigma^{\frac{1}{2}} \mathbf{U}_\Sigma^\top \otimes \mathbf{S}_\Omega^{\frac{1}{2}} \mathbf{U}_\Omega^\top\right) \\
&= \left(\mathbf{U}_\Sigma \mathbf{S}_\Sigma^{\frac{1}{2}} \otimes \mathbf{U}_\Omega \mathbf{S}_\Omega^{\frac{1}{2}}\right)\left(\tilde{\mathbf{C}} \otimes \tilde{\mathbf{R}} + \mathbf{I} \otimes \mathbf{I}\right)\left(\mathbf{S}_\Sigma^{\frac{1}{2}} \mathbf{U}_\Sigma^\top \otimes \mathbf{S}_\Omega^{\frac{1}{2}} \mathbf{U}_\Omega^\top\right) \\
&= \left(\mathbf{U}_\Sigma \mathbf{S}_\Sigma^{\frac{1}{2}} \otimes \mathbf{U}_\Omega \mathbf{S}_\Omega^{\frac{1}{2}}\right) \tilde{\mathbf{K}} \left(\mathbf{S}_\Sigma^{\frac{1}{2}} \mathbf{U}_\Sigma^\top \otimes \mathbf{S}_\Omega^{\frac{1}{2}} \mathbf{U}_\Omega^\top\right),
\end{aligned}
\tag{9}
$$

where $\tilde{\mathbf{C}} \otimes \tilde{\mathbf{R}} + \mathbf{I} \otimes \mathbf{I}$ is the projected covariance matrix $\tilde{\mathbf{K}}$.

At the slight cost of computing the eigenvalue decompositions of $\boldsymbol{\Sigma}$ and $\boldsymbol{\Omega}$, we can now bring the log likelihood in a similar form as for multi-task GP regression with iid noise [2, 4]:

$$
\begin{aligned}
\mathcal{L} = & -\frac{NT}{2}\ln(2\pi) - \frac{1}{2}\ln|\mathbf{K}| - \frac{1}{2}\mathrm{vec}\mathbf{Y}^\top \mathbf{K}^{-1}\mathrm{vec}\mathbf{Y} \\
= & -\frac{NT}{2}\ln(2\pi) - \ln\left|\left(\mathbf{U}_\Sigma \mathbf{S}_\Sigma^{\frac{1}{2}} \otimes \mathbf{U}_\Omega \mathbf{S}_\Omega^{\frac{1}{2}}\right)\tilde{\mathbf{K}}\left(\mathbf{S}_\Sigma^{\frac{1}{2}}\mathbf{U}_\Sigma^\top \otimes \mathbf{S}_\Omega^{\frac{1}{2}}\mathbf{U}_\Omega^\top\right)\right| \\
& - \frac{1}{2}\mathrm{vec}\mathbf{Y}^\top \left[\left(\mathbf{U}_\Sigma \mathbf{S}_\Sigma^{\frac{1}{2}} \otimes \mathbf{U}_\Omega \mathbf{S}_\Omega^{\frac{1}{2}}\right)\tilde{\mathbf{K}}\left(\mathbf{S}_\Sigma^{\frac{1}{2}}\mathbf{U}_\Sigma^\top \otimes \mathbf{S}_\Omega^{\frac{1}{2}}\mathbf{U}_\Omega^\top\right)\right]^{-1}\mathrm{vec}\mathbf{Y} \\
= & -\frac{NT}{2}\ln(2\pi) - \frac{1}{2}\ln|\mathbf{U}_\Sigma \mathbf{S}_\Sigma \mathbf{U}_\Sigma^\top \otimes \mathbf{U}_\Omega \mathbf{S}_\Omega \mathbf{U}_\Omega^\top| - \frac{1}{2}\ln|\tilde{\mathbf{K}}| \\
& - \frac{1}{2}\mathrm{vec}\mathbf{Y}^\top \left(\mathbf{U}_\Sigma \mathbf{S}_\Sigma^{-\frac{1}{2}} \otimes \mathbf{U}_\Omega \mathbf{S}_\Omega^{-\frac{1}{2}}\right)\tilde{\mathbf{K}}^{-1}\left(\mathbf{S}_\Sigma^{-\frac{1}{2}}\mathbf{U}_\Sigma^\top \otimes \mathbf{S}_\Omega^{-\frac{1}{2}}\mathbf{U}_\Omega^\top\right)\mathrm{vec}\mathbf{Y} \\
= & -\frac{NT}{2}\ln(2\pi) - \ln|\mathbf{S}_\Sigma \otimes \mathbf{S}_\Omega| - \frac{1}{2}\ln|\tilde{\mathbf{K}}| \\
& - \frac{1}{2}\left[\left(\mathbf{S}_\Sigma^{-\frac{1}{2}}\mathbf{U}_\Sigma^\top \otimes \mathbf{S}_\Omega^{-\frac{1}{2}}\mathbf{U}_\Omega^\top\right)\mathrm{vec}\mathbf{Y}\right]^\top \tilde{\mathbf{K}}^{-1}\left[\left(\mathbf{S}_\Sigma^{-\frac{1}{2}}\mathbf{U}_\Sigma^\top \otimes \mathbf{S}_\Omega^{-\frac{1}{2}}\mathbf{U}_\Omega^\top\right)\mathrm{vec}\mathbf{Y}\right] \\
= & -\frac{NT}{2}\ln(2\pi) - \ln|\mathbf{S}_\Sigma \otimes \mathbf{S}_\Omega| - \frac{1}{2}\ln|\tilde{\mathbf{K}}| \\
& - \frac{1}{2}\mathrm{vec}\left(\mathbf{S}_\Omega^{-\frac{1}{2}}\mathbf{U}_\Omega^\top \mathbf{Y}\mathbf{U}_\Sigma \mathbf{S}_\Sigma^{-\frac{1}{2}}\right)^\top \tilde{\mathbf{K}}^{-1}\mathrm{vec}\left(\mathbf{S}_\Omega^{-\frac{1}{2}}\mathbf{U}_\Omega^\top \mathbf{Y}\mathbf{U}_\Sigma \mathbf{S}_\Sigma^{-\frac{1}{2}}\right) \\
= & -\frac{NT}{2}\ln(2\pi) - \frac{N}{2}\sum_{i=1}^{T}\ln\mathbf{S}_\Sigma[i,i] - \frac{T}{2}\sum_{j=1}^{N}\ln\mathbf{S}_\Omega[j,j] - \frac{1}{2}\ln|\tilde{\mathbf{K}}| - \frac{1}{2}\mathrm{vec}\tilde{\mathbf{Y}}^\top \tilde{\mathbf{K}}^{-1}\mathrm{vec}\tilde{\mathbf{Y}}
\end{aligned}
\tag{10}
$$

where $\mathbf{S}_\Omega^{-\frac{1}{2}}\mathbf{U}_\Omega^\top \mathbf{Y}\mathbf{U}_\Sigma \mathbf{S}_\Sigma^{-\frac{1}{2}}$ is the projected outcome $\tilde{\mathbf{Y}}$. Computing the eigenvalue decompositions of $\boldsymbol{\Sigma}$ and $\boldsymbol{\Omega}$ takes $O(N^3 + T^3)$ time, and transforming $\mathbf{Y}$ to $\tilde{\mathbf{Y}}$ takes $O(N^2 T + T^2 N)$ time, since $\mathbf{S}_\Omega, \mathbf{S}_\Sigma$ are diagonal.

Along similar lines as discussed for the iid case, we can now efficiently evaluate the marginal log likelihood:

$$
\begin{aligned}
\mathcal{L} = & -\frac{NT}{2}\ln(2\pi) - \frac{N}{2}\sum_{i=1}^{T}\ln\mathbf{S}_\Sigma[i,i] - \frac{T}{2}\sum_{j=1}^{N}\ln\mathbf{S}_\Omega[j,j] - \frac{1}{2}\ln|\tilde{\mathbf{K}}| - \frac{1}{2}\mathrm{vec}\tilde{\mathbf{Y}}^\top \tilde{\mathbf{K}}^{-1}\mathrm{vec}\tilde{\mathbf{Y}} \\
= & -\frac{NT}{2}\ln(2\pi) - \frac{N}{2}\sum_{i=1}^{T}\ln\mathbf{S}_\Sigma[i,i] - \frac{T}{2}\sum_{j=1}^{N}\ln\mathbf{S}_\Omega[j,j] - \frac{1}{2}\ln|\mathbf{S}_{\tilde{\mathbf{C}}} \otimes \mathbf{S}_{\tilde{\mathbf{R}}} + \mathbf{I} \otimes \mathbf{I}| \\
& - \frac{1}{2}\left(\mathrm{vec}\tilde{\mathbf{Y}}\right)^\top \left[\left(\mathbf{U}_{\tilde{\mathbf{C}}} \otimes \mathbf{U}_{\tilde{\mathbf{R}}}\right)\left(\mathbf{S}_{\tilde{\mathbf{C}}} \otimes \mathbf{S}_{\tilde{\mathbf{R}}} + \mathbf{I}\right)\left(\mathbf{U}_{\tilde{\mathbf{C}}} \otimes \mathbf{U}_{\tilde{\mathbf{R}}}\right)^\top\right]^{-1}\mathrm{vec}\tilde{\mathbf{Y}} \\
= & -\frac{NT}{2}\ln(2\pi) - \frac{N}{2}\sum_{i=1}^{T}\ln\mathbf{S}_\Sigma[i,i] - \frac{T}{2}\sum_{j=1}^{N}\ln\mathbf{S}_\Omega[j,j] - \frac{1}{2}\ln|\mathbf{S}_{\tilde{\mathbf{C}}} \otimes \mathbf{S}_{\tilde{\mathbf{R}}} + \mathbf{I} \otimes \mathbf{I}| \\
& - \frac{1}{2}\left[\left(\mathbf{U}_{\tilde{\mathbf{C}}}^\top \otimes \mathbf{U}_{\tilde{\mathbf{R}}}^\top\right)\mathrm{vec}\tilde{\mathbf{Y}}\right]^T \left(\mathbf{S}_{\tilde{\mathbf{C}}} \otimes \mathbf{S}_{\tilde{\mathbf{R}}} + \mathbf{I}\right)^{-1}\left[\left(\mathbf{U}_{\tilde{\mathbf{C}}}^\top \otimes \mathbf{U}_{\tilde{\mathbf{R}}}^\top\right)\mathrm{vec}\tilde{\mathbf{Y}}\right] \\
= & -\frac{NT}{2}\ln(2\pi) - \frac{N}{2}\sum_{i=1}^{T}\ln\mathbf{S}_\Sigma[i,i] - \frac{T}{2}\sum_{j=1}^{N}\ln\mathbf{S}_\Omega[j,j] - \frac{1}{2}\sum_{i=1}^{T}\sum_{j=1}^{N}(\mathbf{S}_{\tilde{\mathbf{C}}}[i,i]\mathbf{S}_{\tilde{\mathbf{R}}}[j,j] + 1) \\
& - \frac{1}{2}\left(\mathbf{U}_{\tilde{\mathbf{R}}}^\top \tilde{\mathbf{Y}}\mathbf{U}_{\tilde{\mathbf{C}}}\right)^\top \left(\mathbf{S}_{\tilde{\mathbf{C}}} \otimes \mathbf{S}_{\tilde{\mathbf{R}}} + \mathbf{I}\right)^{-1}\mathrm{vec}\left(\mathbf{U}_{\tilde{\mathbf{R}}}^\top \tilde{\mathbf{Y}}\mathbf{U}_{\tilde{\mathbf{C}}}\right)
\end{aligned}
$$

$$\tag{11}$$
$$\tag{12}$$

In line (11), we exploit the fact that the eigenvalue decomposition of $\tilde{\mathbf{K}}$ can be recovered by the eigenvalue decompositions of $\tilde{\boldsymbol{\Sigma}}$ and $\tilde{\boldsymbol{\Omega}}$. Computing these decompositions takes $O(N^3 + T^3)$ time, the log determinant can then be computed in $O(NT)$ time and the squared form in $O(N^2 T + NT^2)$ time since $\mathbf{S}_{\tilde{\mathbf{C}}}, \mathbf{S}_{\tilde{\mathbf{R}}}$ are diagonal.

**Efficient gradient evaluation**

$$\begin{aligned}
\frac{\partial}{\partial \theta_R}\mathcal{L} =& -\frac{1}{2}\frac{\partial}{\partial \theta_R}\ln|\mathbf{K}| - \frac{1}{2}\mathrm{vec}\mathbf{Y}^\top\left[\frac{\partial}{\partial \theta_R}\mathbf{K}^{-1}\right]\mathrm{vec}(\mathbf{Y})\\
=& -\frac{1}{2}\frac{\partial}{\partial \theta_R}\ln|\tilde{\mathbf{K}}| - \frac{1}{2}\mathrm{vec}\tilde{\mathbf{Y}}^\top\left[\frac{\partial}{\partial \theta_R}\tilde{\mathbf{K}}^{-1}\right]\mathrm{vec}(\tilde{\mathbf{Y}})\\
=& -\frac{1}{2}\mathrm{Tr}\left[\tilde{\mathbf{K}}^{-1}\frac{\partial}{\partial \theta_R}\left(\tilde{\mathbf{C}}\otimes\tilde{\mathbf{R}}+\mathbf{I}\otimes\mathbf{I}\right)\right] + \frac{1}{2}\mathrm{vec}\tilde{\mathbf{Y}}^\top\left[\tilde{\mathbf{K}}^{-1}\left(\frac{\partial}{\partial \theta_R}\left(\tilde{\mathbf{C}}\otimes\tilde{\mathbf{R}}+\mathbf{I}\otimes\mathbf{I}\right)\right)\tilde{\mathbf{K}}^{-1}\right]\mathrm{vec}(\tilde{\mathbf{Y}})\\
=& -\frac{1}{2}\mathrm{Tr}\left[(\mathbf{U}_{\tilde{\mathbf{C}}}\otimes\mathbf{U}_{\tilde{\mathbf{R}}})(\mathbf{S}_{\tilde{\mathbf{C}}}\otimes\mathbf{S}_{\tilde{\mathbf{R}}}+\mathbf{I})^{-1}(\mathbf{U}_{\tilde{\mathbf{C}}}^\top\otimes\mathbf{U}_{\tilde{\mathbf{R}}}^\top)\left(\tilde{\mathbf{C}}\otimes\frac{\partial}{\partial \theta_R}\tilde{\mathbf{R}}\right)\right]\\
&+\frac{1}{2}\left(\tilde{\mathbf{K}}^{-1}\mathrm{vec}\tilde{\mathbf{Y}}\right)^\top\left(\tilde{\mathbf{C}}\otimes\frac{\partial}{\partial \theta_R}\tilde{\mathbf{R}}\right)\left(\tilde{\mathbf{K}}^{-1}\mathrm{vec}\tilde{\mathbf{Y}}\right)\\
=& -\frac{1}{2}\mathrm{Tr}\left[(\mathbf{U}_{\tilde{\mathbf{C}}}\otimes\mathbf{U}_{\tilde{\mathbf{R}}})(\mathbf{S}_{\tilde{\mathbf{C}}}\otimes\mathbf{S}_{\tilde{\mathbf{R}}}+\mathbf{I})^{-1}(\mathbf{U}_{\tilde{\mathbf{C}}}^\top\otimes\mathbf{U}_{\tilde{\mathbf{R}}}^\top)\left(\tilde{\mathbf{C}}\otimes\frac{\partial}{\partial \theta_R}\tilde{\mathbf{R}}\right)\right]\\
&+\frac{1}{2}\left((\mathbf{U}_{\tilde{\mathbf{C}}}\otimes\mathbf{U}_{\tilde{\mathbf{R}}})(\mathbf{S}_{\tilde{\mathbf{C}}}\otimes\mathbf{S}_{\tilde{\mathbf{R}}}+\mathbf{I})^{-1}(\mathbf{U}_{\tilde{\mathbf{C}}}^\top\otimes\mathbf{U}_{\tilde{\mathbf{R}}}^\top)\mathrm{vec}\tilde{\mathbf{Y}}\right)^\top\left(\tilde{\mathbf{C}}\otimes\frac{\partial}{\partial \theta_R}\tilde{\mathbf{R}}\right)\\
&\qquad\qquad\qquad\qquad\left((\mathbf{U}_{\tilde{\mathbf{C}}}\otimes\mathbf{U}_{\tilde{\mathbf{R}}})(\mathbf{S}_{\tilde{\mathbf{C}}}\otimes\mathbf{S}_{\tilde{\mathbf{R}}}+\mathbf{I})^{-1}(\mathbf{U}_{\tilde{\mathbf{C}}}^\top\otimes\mathbf{U}_{\tilde{\mathbf{R}}}^\top)\mathrm{vec}\tilde{\mathbf{Y}}\right)\\
=& -\frac{1}{2}\mathrm{Tr}\left[(\mathbf{S}_{\tilde{\mathbf{C}}}\otimes\mathbf{S}_{\tilde{\mathbf{R}}}+\mathbf{I})^{-1}\left(\mathbf{U}_{\tilde{\mathbf{C}}}^\top\tilde{\mathbf{C}}\mathbf{U}_{\tilde{\mathbf{C}}}\otimes\mathbf{U}_{\tilde{\mathbf{R}}}^\top\frac{\partial}{\partial \theta_R}\tilde{\mathbf{R}}\mathbf{U}_{\tilde{\mathbf{R}}}\right)\right]\\
&+\frac{1}{2}\left((\mathbf{U}_{\tilde{\mathbf{C}}}\otimes\mathbf{U}_{\tilde{\mathbf{R}}})\mathrm{vec}\hat{\mathbf{Y}}\right)^\top\left(\tilde{\mathbf{C}}\otimes\frac{\partial}{\partial \theta_R}\tilde{\mathbf{R}}\right)\left((\mathbf{U}_{\tilde{\mathbf{C}}}\otimes\mathbf{U}_{\tilde{\mathbf{R}}})\mathrm{vec}\hat{\mathbf{Y}}\right)\\
=& -\frac{1}{2}\mathrm{Tr}\left[(\mathbf{S}_{\tilde{\mathbf{C}}}\otimes\mathbf{S}_{\tilde{\mathbf{R}}}+\mathbf{I})^{-1}\left(\mathbf{U}_{\tilde{\mathbf{C}}}^\top\tilde{\mathbf{C}}\mathbf{U}_{\tilde{\mathbf{C}}}\otimes\mathbf{U}_{\tilde{\mathbf{R}}}^\top\frac{\partial}{\partial \theta_R}\tilde{\mathbf{R}}\mathbf{U}_{\tilde{\mathbf{R}}}\right)\right]\\
&+\frac{1}{2}\mathrm{vec}\hat{\mathbf{Y}}^\top\left(\mathbf{U}_{\tilde{\mathbf{C}}}^\top\tilde{\mathbf{C}}\otimes\mathbf{U}_{\tilde{\mathbf{R}}}^\top\frac{\partial}{\partial \theta_R}\tilde{\mathbf{R}}\right)\mathrm{vec}\left(\mathbf{U}_{\tilde{\mathbf{R}}}\hat{\mathbf{Y}}\mathbf{U}_{\tilde{\mathbf{C}}}^\top\right)\\
=& -\frac{1}{2}\mathrm{Tr}\left[(\mathbf{S}_{\tilde{\mathbf{C}}}\otimes\mathbf{S}_{\tilde{\mathbf{R}}}+\mathbf{I})^{-1}\left(\mathbf{S}_{\tilde{\mathbf{C}}}\otimes\mathbf{U}_{\tilde{\mathbf{R}}}^\top\frac{\partial}{\partial \theta_R}\tilde{\mathbf{R}}\mathbf{U}_{\tilde{\mathbf{R}}}\right)\right]\\
&+\frac{1}{2}\left(\mathrm{vec}\hat{\mathbf{Y}}\right)^\top\mathrm{vec}\left(\left(\mathbf{U}_{\tilde{\mathbf{R}}}^\top\frac{\partial}{\partial \theta_R}\tilde{\mathbf{R}}\right)\left(\mathbf{U}_{\tilde{\mathbf{R}}}\hat{\mathbf{Y}}\mathbf{U}_{\tilde{\mathbf{C}}}^\top\right)\left(\mathbf{U}_{\tilde{\mathbf{C}}}^\top\tilde{\mathbf{C}}\right)^\top\right)\\
=& -\frac{1}{2}\mathrm{diag}\left((\mathbf{S}_{\tilde{\mathbf{C}}}\otimes\mathbf{S}_{\tilde{\mathbf{R}}}+\mathbf{I})^{-1}\right)^\top\mathrm{diag}\left(\mathbf{S}_{\tilde{\mathbf{C}}}\otimes\mathbf{U}_{\tilde{\mathbf{R}}}^\top\frac{\partial}{\partial \theta_R}\tilde{\mathbf{R}}\mathbf{U}_{\tilde{\mathbf{R}}}\right)\\
&+\frac{1}{2}\left(\mathrm{vec}\hat{\mathbf{Y}}\right)^\top\mathrm{vec}\left(\mathbf{U}_{\tilde{\mathbf{R}}}^\top\left(\frac{\partial}{\partial \theta_R}\tilde{\mathbf{R}}\right)\mathbf{U}_{\tilde{\mathbf{R}}}\hat{\mathbf{Y}}\mathbf{S}_{\tilde{\mathbf{C}}}\right),\qquad\qquad\text{(13)}
\end{aligned}$$

where $(\mathbf{S}_{\tilde{\mathbf{C}}}\otimes\mathbf{S}_{\tilde{\mathbf{R}}}+\mathbf{I})^{-1}\left(\mathbf{U}_{\tilde{\mathbf{C}}}^\top\otimes\mathbf{U}_{\tilde{\mathbf{R}}}^\top\right)\mathrm{vec}\tilde{\mathbf{Y}} = (\mathbf{S}_{\tilde{\mathbf{C}}}\otimes\mathbf{S}_{\tilde{\mathbf{R}}}+\mathbf{I})^{-1}\mathrm{vec}\left(\mathbf{U}_{\tilde{\mathbf{R}}}^\top\tilde{\mathbf{Y}}\mathbf{U}_{\tilde{\mathbf{C}}}\right)$ is $\hat{\mathbf{Y}}$.

**Efficient prediction**

$$
\begin{aligned}
\mathbf{m}^* &= \left(\mathbf{C} \otimes \mathbf{R}^*\right)\left(\mathbf{C} \otimes \mathbf{R} + \boldsymbol{\Sigma} \otimes \mathbf{I}\right)^{-1}\operatorname{vec}\mathbf{Y}\\
&= \left(\mathbf{C} \otimes \mathbf{R}^*\right)\left[\left(\mathbf{U}_\Sigma \mathbf{S}_\Sigma^{\frac{1}{2}} \otimes \mathbf{U}_\Omega \mathbf{S}_\Omega^{\frac{1}{2}}\right)\tilde{\mathbf{K}}\left(\mathbf{S}_\Sigma^{\frac{1}{2}}\mathbf{U}_\Sigma^\top \otimes \mathbf{S}_\Omega^{\frac{1}{2}}\mathbf{U}_\Omega^\top\right)\right]^{-1}\operatorname{vec}\mathbf{Y}\\
&= \left(\mathbf{C} \otimes \mathbf{R}^*\right)\left(\mathbf{U}_\Sigma \mathbf{S}_\Sigma^{-\frac{1}{2}} \otimes \mathbf{U}_\Omega \mathbf{S}_\Omega^{-\frac{1}{2}}\right)\tilde{\mathbf{K}}^{-1}\left(\mathbf{S}_\Sigma^{-\frac{1}{2}}\mathbf{U}_\Sigma^\top \otimes \mathbf{S}_\Omega^{-\frac{1}{2}}\mathbf{U}_\Omega^\top\right)\operatorname{vec}\mathbf{Y}\\
&= \left(\mathbf{C}\mathbf{U}_\Sigma \mathbf{S}_\Sigma^{-\frac{1}{2}} \otimes \mathbf{R}^*\mathbf{U}_\Omega \mathbf{S}_\Omega^{-\frac{1}{2}}\right)\tilde{\mathbf{K}}^{-1}\operatorname{vec}\left(\left(\mathbf{S}_\Omega^{-\frac{1}{2}}\mathbf{U}_\Omega^\top\right)\mathbf{Y}\left(\mathbf{U}_\Sigma \mathbf{S}_\Sigma^{-\frac{1}{2}}\right)\right)\\
&= \left(\mathbf{C}\mathbf{U}_\Sigma \mathbf{S}_\Sigma^{-\frac{1}{2}} \otimes \mathbf{R}^*\mathbf{U}_\Omega \mathbf{S}_\Omega^{-\frac{1}{2}}\right)\left(\tilde{\mathbf{K}}^{-1}\operatorname{vec}\tilde{\mathbf{Y}}\right)\\
&= \left(\mathbf{C}\mathbf{U}_\Sigma \mathbf{S}_\Sigma^{-\frac{1}{2}} \otimes \mathbf{R}^*\mathbf{U}_\Omega \mathbf{S}_\Omega^{-\frac{1}{2}}\right)\operatorname{vec}\left(\mathbf{U}_{\tilde{\mathbf{R}}}\hat{\mathbf{Y}}\mathbf{U}_{\tilde{\mathbf{C}}}^\top\right)\\
&= \operatorname{vec}\left(\left(\mathbf{R}^*\mathbf{U}_\Omega \mathbf{S}_\Omega^{-\frac{1}{2}}\right)\left(\mathbf{U}_{\tilde{\mathbf{R}}}\hat{\mathbf{Y}}\mathbf{U}_{\tilde{\mathbf{C}}}^\top\right)\left(\mathbf{C}\mathbf{U}_\Sigma \mathbf{S}_\Sigma^{-\frac{1}{2}}\right)^\top\right)\\
&= \operatorname{vec}\left(\mathbf{R}^*\mathbf{U}_\Omega \mathbf{S}_\Omega^{-\frac{1}{2}}\mathbf{U}_{\tilde{\mathbf{R}}}\hat{\mathbf{Y}}\mathbf{U}_{\tilde{\mathbf{C}}}^\top\mathbf{S}_\Sigma^{-\frac{1}{2}}\mathbf{U}_\Sigma^\top\mathbf{C}^\top\right)
\end{aligned}
\tag{14}
$$