[Reviews · NeurIPS 2013]

Submitted by Assigned_Reviewer_1

Update after reading the authors' rebuttal:

Very good paper but I strongly recommend the authors to consider all the suggestions below that will improve the readability of the paper.

=====

1. Summary:

This paper presents a multi-task Gaussian process regression approach where the covariance of the main process (signal) decomposes as a product of a covariance between tasks and a covariance between inputs (sample covariance). It is assumed that all the training outputs are observed at all inputs, which leads to a Kronecker product covariance.

Noisy observations are modeled via a structured process and this is the main contribution of the paper. While previous work on multi-task GP approaches with Kronecker covariances has considered iid noise in order to carry out efficient computations, this paper shows that it is possible to consider a noise process with Kronecker structure, while maintaining efficient computations. In other words, as in the iid noise case, one never has to compute a Kronecker product and hence computations are O(N^3 + T^3) instead of O(N^3T^3). This is achieved by whitening the noise process and projecting the (noiseless) covariance of the system into the eigen-basis of the noise covariance (scaled by the eigenvalues).

Their experiments show that the proposed structured-noise multi-task GP approach outperforms the baseline iid-noise multi-task GP method and independent GPs on synthetic data and real applications.



2. Quality

The paper seems technically sound once one realizes how to fix the notational inconsistencies between section 2 and section 3 (please see item 4 below regarding Clarity). The main claim that one can have structured noise in multi-task GP models with product covariances while still maintaining computational efficiency (compared to the naive approach) is well supported with the mathematical derivations in section 3 and with the experimental results in section 4.

With respect to the results, although it is obvious that the naive approach would be much worse in terms of computation compared to the efficient approach, it is still helpful to see the comparison in Figure 1 so one can take into consideration the possible overhead in implementing the GP-KS method. However, there are a few deficiencies that need to be pointed out:

(a) There is a rank-1 parameterization of C and \Sigma. However, it is unclear how the parameter \sigma in line 246 was set. This parameter is important as it allows that the algorithm can actually run but also that one does not over-smooth the process leading to poor generalization performance.

(b) It is completely unclear what \Omega is actually used (please see Clarity below). This should be explicitly said in the experiments. From section 2, it can be inferred that \Omega = I_{NN}. But the reader should not be guessing about something that must be explicit.

(c) Line 328: "Large-scale prediction .." : 123 tasks and 218 samples are very far from what can be considered as large-scale.

(d) In the experiments regarding the prediction of gene expression in yeast, it looks like the preprocessing and filtering of the dataset does favor the proposed method (which may have problems with identifiability) as genes with low signal and low noise are discarded. The authors should provide comments on this.

(e) The authors have not analyzed possible weaknesses of their method. In particular, interpreting the results in Figure 3 is a bit misleading as it seems that their method has high levels of unidentifiability. Why is it possible to interpret the results? There may be completely different qualitative solutions that lead to similar quantitative performance. Is identifiability an issue during optimization?

(f) Multi-task settings usually compare to a single GP for all tasks (i.e. pool GP). This baseline is missing.

3. Clarity

In terms of language use, the paper is relatively well written. However, there are quite a few notational inconsistencies that may push this paper below the threshold. For example:

(a) The same symbol (k) in Equation 1 is used for both covariance (sample and task) functions and this is completely inconsistent with the following notation in the paper (C, R). On the same equation, is this really the covariance between the noisy outputs y or should it be between their corresponding noiseless latent functions?

(b) Sometimes C_{TT}, R_{NN}, etc are used and other times C, T, are used

(c) This is inconsistency is crucial to understand the paper: I_{NN} is used in Equation 3 but then
\Omega is introduced in section 3 without explaining what it refers to. Is \Omega = I_{NN}?

(d) K is undefined before being used in Equation 6

(e) Equation 9 does not make sense. It goes from a vector on the line above to a matrix. Is there a Vec operator missing?


4. Originality

The approach to multi-task GP regression differs from most previous work in the way the noise process is modeled (structured noise compared to idd noise) while maintaining efficiency during inference. These types of processes have been considered before for example by Zhang (2007) but efficient computations were not explicitly done when considering the specific case of two processes.

Without committing to a specific approach for flexible multi-task models, it seems necessary to at least mention how this works compare to the Gaussian process regression network framework in [6].

Additionally, sec 2.2 is obvious and should be omitted. The case C = \Sigma leads to C \otimes (R+I), which is the same proof shown in previous work [1].

5. Significance

The contribution of this paper is relatively significant in that it shows that it is possible to do efficient computation in these types of models when a sum of two Kronecker products is present. This can be exploited in scenarios different to the regression setting. However, in terms of the original motivation, i.e. multi-task regression, there are other more flexible models [6] for which inference is still better than the naive approach (N^3 T^3).

6. Additional comments

(a) Abstract, lines 18-20: This is not true.


References:

Hao Zhang. Maximum-likelihood estimation for multivariate spatial linear coregionalization models. Environmetrics, 18(2):125–139, 2007
Summary: This paper presents a novel approach to multi-task GP regression where the noise process is structured. It shows that inference can be carried out more efficiently compared to the naive approach. The experimental results show the proposed approach is better than previous work that used iid noise. Quite a few notational problems need to be fixed if this paper is to be published.

Submitted by Assigned_Reviewer_2

Overview
==
The method proposes a sum of kronecker kernels for GP regression. The idea is that one kernel represents signal (ie. is dependent on the inputs), and the other represents some structured noise. An efficient inference scheme is derived, and some convincing experiments o statistical genetics are presented.

A strong paper with a clear flow. Some details could be clarified, and there is some slopiness in the notation, but this could easily be overcome in the rebuttal stage.


Introduction
--
A neat introduction. I like the approach through Bonilla and Williams' result. I wonder if you could expand this slightly: why does the prediction reduce to independent models?


Section 2
--

minor quibble: definition of vec Y is a little sloppy. consider using
\vec {\mathbf Y} = (y_1^\top \ldots y_T^\top)^\top

line 124: This is the same as a GPLVM with a _linear kernel_. I think this is going to confuse some readers, suggest you either expand or omit.

line 136: Y*_{n,t} = ... this is not Y*, but the mean prediction of Y*. perhaps you should denote it M*_{n,t}?

eq. (5). I like this derivation, but it took me a little while to follow ( I had to look up the rules for kroneker multiplication). Perhaps you could expand some of the steps in an appendix?

Section 3
--
This is the heart of the paper, and the main contribution I feel. But you've not introduced \Omega!

It would be good to know why K_tilde is easier to deal with than K. Is it smaller (fewer eigenvalues) or is the Kronecker of the identity easy to deal with?


Section 4
==
The simulation section is great. It's clear that your proposed method is working well.

minor quibble: line 269 -- I'm not sure that drastic is the correct term! Perhaps 'dramatic', or 'significant'.

line 360. Your discussion isn't so clear to me. I can see that your model worked in some sense, in that the recovered noise covariance has structure, and clearly it's hard to come up with concrete validation with out a gold standard, but it's not clear what you're demonstrating. How are the conditions organised in the covariance matrices of fig 3? I guess one condition is the first block of the matrix, and the other condition is the next block? More explanation required, please.


Pros
==
- A very well written paper (with a few exception, above) which flows well and is readily understood.
- Simple idea, but effective. Novel to my knowledge.


Cons
==
- The application might be of interest to a limited portion of the NIPS community


Summary: Nice paper, needs a little clarification in places before publication but otherwise good.

Submitted by Assigned_Reviewer_5

This paper discusses GP regression for the multi-task case, and specifically whether one should allow correlations between tasks in the noise/residuals. It shows that such correlations can be dealt with efficiently, basically by rotating in task space so that one gets back to uncorrelated noise. Applications to biological problems with low-rank/factor-analysis type task correlations show improvements over the uncorrelated noise case.
Presentation is good but confusing in parts - e.g. in Sec 3 an Omega suddenly appears which up to there was an identity matrix, presumably to allow correlations in noise across data points? Motivation for this seems unclear. "ln" missing in first line of (7)? (d,d') on p1 should be called (t,t') for consistency with notation later. The GP-KP and GP-KS acronyms are easy to mix up and the authors themselves get muddled (they also have GS-KP and GS-KS).
Summary: Nice paper on allowing correlated (between tasks) noise in multi-task GP regression, dealt with efficiently by "rotating away" these correlations. Applications to biological data seem convincing.

Submitted by Assigned_Reviewer_7

In this work, efficient inference is presented for multi-task GPs having different signal and noise structure (inter-task covariance).

This work is well-written and organized. Its main contribution is to emphasize the importance of noise in multi-task GP prediction: When noise and signal have the same inter-task covariance, or noise is not present, a multi-task GP produces the same mean posterior as independent GPs. This had been mentioned before in [1], but not emphasized enough.

Efficient inference for the "useful" case in which both structures are different is provided. This is rather straightforward given existing literature on the topic.

The paper could be improved by:

- Providing a reasonable example in which the "useful" case arises naturally. An attempt at this is made when talking about "unobserved causal features". First, I would like to point out that the word "causal" might be unfortunate here. The reasoning applies equally as long as the feature is a useful input for prediction, no matter whether it is a cause, a consequence, or none. Second, the explanation about how Y_hidden is generated is missing. If it was generated just as Y_obs is generated, it would have the same structure. The authors imply that this is not the case, but it would be interesting to mention a natural process with such behavior.

- Giving more detail about why (7) is a more efficient version. Matrices that require inversion have the same size, some readers might not be familiar with the properties of Kronecker products of diagonal matrices.

- The equation inside the second paragraph of Sec. 2.2 using vec(Y) is dimensionally inconsistent.

- Omega seems to be used in Sec. 3 as a placeholder for the previously homoscedastic noise, but this is not explained.

- If tasks turn out to be independent, this model restricts them all to have the same signal power (according to the proposed diagonal plus rank one matrix). This might be unrealistic.
Summary: This work emphasizes the influence of noise structure in making multi-task GPs useful. Derivations are quite straightforward but result in a useful model, which is a variation of previously existing multi-task GPs (noise and signal have different inter-task covariances).
Author Feedback

Author rebuttal: All Reviewers:
We thank all reviewers for pointing out that the definition of Omega is missing. In Section 3, we show how efficient inference can be done for an arbitrary sum of two kronecker products, while the application to multi task prediction is mainly concerned with the special case Omega=I_{NN}. We will clarify that in the final submission.

Following up the suggestions of Reviewer 2 and 4, we will also provide a more comprehensive derivation of the equations in the Appendix.

Reviewer 1:
How the parameter sigma in line 246 was set.
-All hyperparameters, including sigma, were obtained by gradient-based optimization of the marginal likelihood.

In the experiments regarding the prediction of gene expression in yeast, it looks like the preprocessing and filtering of the dataset does favor the proposed method (which may have problems with identifiability) as genes with low signal and low noise are discarded.
-In our experiments, we followed the design choice of [1,6] and employed a common noise level sigma for all tasks. However, it is possible to consider one noise level for each task, which would be appropriate for larger number of tasks with variable signal-to-noise ratio.

In particular, interpreting the results in Figure 3 is a bit misleading as it seems that their method has high levels of unidentifiability.
-It is true that our method, as other multitask approaches, is susceptible to local optima. To mitigate the effect of local optima for both prediction and interpretation, we used multiple random restarts and selected the solution with the best out-of-sample likelihood, as described in the Section 4.
For the yeast dataset in particular, we repeated the training 10 times, and computed the mean latent factors and its standard errors: 0.2103+/- 0.0088 (averaged over all latent factors, over the ten best runs selected by out-of-sample likelihood). Moreover, the observed differences were too small to detect by eye. Thus, we believe that our interpretation is valid.

Multi-task settings usually compare to a single GP for all tasks (i.e. pool GP).
-We ran pool GP on the Arabidopsis data, however the method was outperformed by all other competitors with ease (Flowering: 0.0512, Life cycle:0.1051, Maturation:0.0466, Reproduction:0.0488). We would also like to note that both Kronecker models have the pool GP in the space of possible solutions (X_c,X_sigma-->0).

Sec 2.2 is obvious and should be omitted. The case C = \Sigma leads to C \otimes (R+I), which is the same proof shown in previous work [1].
-We agree that the proof in Section 2.2 can be shortened. However, as also pointed out by Reviewer 2, we found the insight that multitask learning cancels when C=\Sigma noteworthy.

We will improve the notation and correct the details as suggested. We will add a more careful comparison to [6] and add recent results from the geostatistics literature (Zhang 2007).
Reviewer 2
I like the approach through Bonilla and Williams' result. I wonder if you could expand this slightly: why does the prediction reduce to independent models?
In the noiseless scenario, the GP mean prediction has the following form:
M*_{n,t}=kron(C,R*)kron(C,R)^(-1)vec(Y)=kron(C,R*)kron(C^(-1),R^(-1))vec(Y)=kron(C*C^(-1),R* * R^(-1))vec(Y) = kron(I,R* * R)vec(Y), and is thus independent of C (see [1]).

It would be good to know why K_tilde is easier to deal with than K. Is it smaller or is the Kronecker of the identity easy to deal with?
-We exploit the fact that [\kron(C,R) + I]^{-1}=[\kron(U_C,U_R)^T (\kron(S_C,S_R)+I)+\kron(U_C,_UR)]^{-1}=
\kron(U_C,U_R)^T(\kron(S_C,S_R) + I)^{-1}\kron(U_C,_UR), where (\kron(S_C,S_R)+I) is diagonal.

line 360. How are the conditions organised in the covariance matrices of fig 3? I guess one condition is the first block of the matrix, and the other condition is the next block?
-The covariance matrices are between the different tasks, while the different conditions are between the samples.
We obtained the ordering by hierarchical clustering between the phenotypes. One can observe that a) rank-1 approximations are sufficient to capture the main trends of the empirical covariance matrix and b) signal and noise covariance matrices reflect different processes, illustrating the benefits from structured noise. We will clarify the description.
Reviewer 3:
We will refine the notation as suggested.
Reviewer 4:
Causality vs. Predictability
-We fully agree with the author that a feature need not to be causal for being predictive and will generalize the description of the simulations to account for that. The simulation by itself does not depend on the assumption that the generated features are causal.

The explanation about how Y_hidden is generated is missing. If it was generated just as Y_obs is generated, it would have the same structure.
-We used different rescalings for Y_hidden (r_hidden) and Y_obs (r_obs) to obtain different task-task covariance matrices (C=r_obs*r_obs^T, \Sigma=r_hidden*r_hidden^T).

The authors imply that this is not the case, but it would be interesting to mention a natural process with such behavior.
-In microarray experiments, gene expression levels are often influenced by genetic factors (observed process) and confounding factors such as batch effects (unobserved process). We mention these examples of natural processes in the final submission.

The equation inside the second paragraph of Sec. 2.2 using vec(Y) is dimensionally inconsistent.
-The equation should be N log|1/N Y^T(R+I)^(-1)Y|.

If tasks turn out to be independent, this model restricts them all to have the same signal power.
-In principle one could introduce a separate noise variance for each target dimension. We chose to use a single noise variance sigma over all tasks as done in [1,6] and many other multitask approaches. However, our efficient inference scheme would also apply to instances with variable noise levels.